# A gravity-driven sintering method to fabricate geometrically complex compact piezoceramics

Yao Shan[1], Shiyuan Liu[1], Biao Wang[1], Ying Hong[1], Chao Zhang[2], C. W. Lim[3], Guangzu Zhang[2] & Zhengbao Yang[1,4 ✉]

Highly compact and geometrically complex piezoceramics are required by a variety of electromechanical devices owing to their outstanding piezoelectricity, mechanical stability and extended application scenarios. 3D printing is currently the mainstream technology for fabricating geometrically complex piezoceramic components. However, it is hard to print piezoceramics in a curve shape while also keeping its compactness due to restrictions on the ceramic loading and the viscosity of feedstocks. Here, we report a gravity-driven sintering (GDS) process to directly fabricate curved and compact piezoceramics by exploiting gravitational force and high-temperature viscous behavior of sintering ceramic specimens. The sintered lead zirconate titanate (PZT) ceramics possess curve geometries that can be facilely tuned via the initial mechanical boundary design, and exhibit high piezoelectric properties comparable to those of conventional-sintered compact PZT ($d_{33} = 595$ pC/N). In contrast to 3D printing technology, our GDS process is suitable for scale-up production and low-cost production of piezoceramics with diverse curved surfaces. Our GDS strategy is an universal and facile route to fabricate curved piezoceramics and other functional ceramics with no compromise of their functionalities.

[1] Department of Mechanical Engineering, City University of Hong Kong, Hong Kong, China. [2] School of Optical and Electronic Information, Engineering Research Center for Functional Ceramic MOE and Wuhan National Laboratory for Optoelectronics, Huazhong University of Science and Technology, Wuhan 430074, China. [3] Department of Architecture and Civil Engineering, City University of Hong Kong, Hong Kong, China. [4] City University of Hong Kong Shenzhen Research Institute, Shenzhen 518057, China. ✉email: zb.yang@cityu.edu.hk

Piezoceramics as a typical type of functional ceramics have been widely used in electronic and energy devices, thanks to their high electromechanical coupling effect, stable mechanical properties, and low cost[1–4]. Tremendous attention has been paid to improving the properties of piezoceramics, including piezoelectricity[5,6], transparency[7,8], and thermal stability[9,10]. Although the superiority of piezoceramics in constructing functional devices has been demonstrated, the use of piezoceramics in complex geometries for various electronic devices remains largely unexplored. The dilemma between geometrical complexity and piezoelectricity of piezoceramic-based devices originates from the inherent trade-off between processing compatibilities and structural compactness of functional ceramics[11]. This raises the question of whether a fabrication strategy can be developed that would allow piezoceramics to simultaneously possess complex geometry and high piezoelectric performance.

Piezoceramics are normally formed into dense bodies starting from green bodies via the sintering process in which the form of green bodies dictates the final shapes of the sintered structures. This fabrication process has been keeping unchanged in the industry since the first piezoceramics synthesis that took place in 1940s[12]. The shape of piezoceramic green bodies has been widely formed via traditional ceramic shaping techniques[13] such as injection molding and die pressing. However, the as-formed green bodies are brittle, and thus challenging to be processed, which results in the geometry simplicity of resultant piezoceramic articles[14].

Substantial effort has been devoted to the development of processing strategies for fabricating piezoceramic architectures with complex shapes. Shaping the green parts and processing the sintered bodies of piezoceramics are two possible approaches to forming the geometrical complexity. 3D printing techniques that allow for forming geometrically complex architectures have opened a promising way for shaping ceramic green-body[11,15], such as slurry-based (e.g., stereolithography (SL)[16,17] and its derivatives[18–22]), powder-based (e.g., selective laser sintering (SLS)[23,24], and electric poling-assisted additive manufacturing (EPAM)[25]) and bulk solid-based (e.g., fused deposition modeling (FDM)[26–29]) methods. However, in the 3D printing process, because the processibility of feedstocks comprised of ceramic constituents and organic binders are linked to the ceramic loading, a compromise is always reached in 3D-printed ceramics where either one of the properties is set aside. These contradicting properties predestine it to be impossible to print geometrically complex ceramic green bodies in a high volume of ceramic constituents and good compactness[30]. For example, for slurry-based ceramic 3D printing techniques, the complex shapes of the ceramic green parts are often at the expense of ceramic loading (up to 60 vol%)[30]. Similarly, for fabricating piezoceramics, powder-based and bulk solid-based 3D printing methods usually require that the sintered bodies sacrifice mechanical and piezoelectric properties for complex geometries. Furthermore, 3D printing methods normally produce only one specimen at a time and is not suitable for large-scale production.

Other attempts to fabricate curved piezoceramics by directly processing ceramic sintered bodies have been reported, which rely on the mismatch of CTE (coefficient of thermal expansion)[31–34] or mechanical properties (prestressed methods) of different layers[35,36]. However, both the CTE-based and mechanically prestressed methods can only provide simple arc-shaped laminates within a small curvature. Furthermore, the properties of the processed samples are limited and strongly dependent on the difference between the properties of the piezoceramic layer and bonded layers (e.g., metal layers and binder layers). The specific laminate structures used in the thermal or prestress post-processing methods limit their generality as well as the ability to form complex 3D geometries (e.g., doubly-curved configurations and large-curvature geometries).

From a physical perspective, the fundamental of forming a complex-shape piezoceramic sintered body is to generate desired deformations at appropriate segments, which always associates with the break and reestablishment of force equilibrium. However, piezoceramic sintered bodies are brittle solids and thus it is challenging to generate sufficiently large deformation under external loads. Therefore, strategies for fabricating curved piezoceramics after sintering like CTE-based and mechanically prestressed methods suffer from limited curvature and shapes. As for the 3D printing technology before sintering, it compromises the piezoelectric and mechanical properties of the sintered bodies to attain the capability of forming designed curved piezoceramic green bodies. These problems and limitations of processing piezoceramics before and after sintering motivate us to consider shaping the piezoceramics during sintering. It has been reported that ceramic sintering body exhibits viscous behavior[37–41], which usually causes a dimensional change of the semifinal specimen during sintering[42–45]. Thus, we take advantage of this unique phenomenon to generate desired deformations on piezoceramic sintering bodies with the assistance of gravitational force.

Here, we report a gravity-driven sintering (GDS) process to fabricate compact piezoceramics in complex 3D configurations. The viscous behavior of piezoceramic green bodies at high temperatures is used to effectively form curved geometries for sintered bodies. We succeed in fabricating lead zirconate titanate (PZT) ceramics that possess high compactness and a curved geometry simultaneously. The GDS strategy shows great potential in facilely fabricating geometrically complex piezoceramics without sacrificing functionalities.

## Results

**Fabrication strategy of the GDS process**. The GDS strategy is shown in Fig. 1a (see Methods for a detailed parameter setup). To build a system that activates the effect of gravitational force during the sintering process, we place a pressed green compact of PZT precursor powders on two supporting alumina platforms that possess low thermal expansion and high thermal conductivity, benefiting for forming a stable high-temperature system in the furnace. In a typical GDS process, the state of the sintering body transforms from solid to quasi-liquid as temperature rises. As shown in Fig. 1b, in the early stage of the GDS process, the gravitational force is balanced by shear force over the cross section and the internal shear force. In the second stage, gravitational force begins to govern the deformation, where force equilibrium sustained in the initial solid-state is broken owing to the viscous properties of the quasi-liquid sintering body (see the second schematic from the left in Fig. 1b). With continuous deformation, a new equilibrium is established and gravity is balanced gradually by the evolving internal shear force[46]. In the final stage, the compact PZT ceramic sintered body within a curved geometry is obtained due to the thermally and mechanically stable system (Fig. 1c). To better understand the technical merits of our GDS process, we compare the processing methods for producing curved and compact piezoceramics in Table 1, where the GDS method is highlighted by its ability to form complex-shaped and compact piezoceramics with high, enhanced piezoelectricity.

**Characterization of GDS-fabricated piezoceramics**. To visualize the deformation discussed above, we record the configuration evolution of the sintering body during the in situ GDS process and the temperature profile, which is shown in Fig. 2a. One can see that the geometry of the green compact remains similar to the

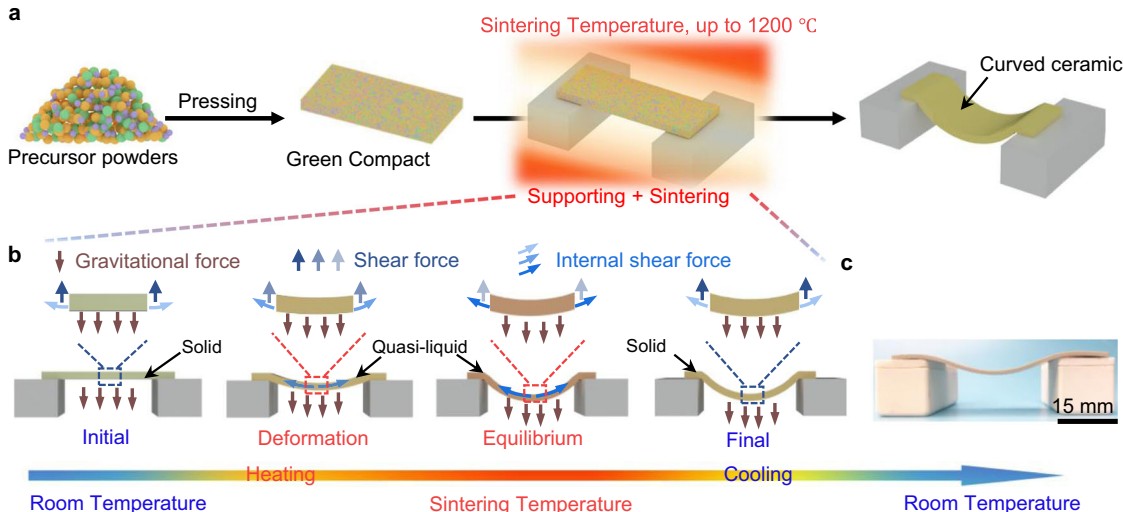

**Fig. 1 Gravity-driven sintering (GDS) process and setup for the ceramic synthesis. a** Schematic of the GDS synthesis process, in which the pressed green compact is placed on two alumina supports and sintered into a dense ceramic. **b** Schematic of the mechanism of the GDS process, in which the force equilibrium in the initial stage is broken in the deformation stage due to the quasi-liquid state of the sintering specimen under high temperature. In the equilibrium stage, a new force equilibrium forms and the curved configuration maintains to the final stage. **c** Photograph of the fabricated PZT ceramic in a curved geometry.

**Table 1 Comparison of processing methods for fabricating compact piezoceramics with complex geometries by the GDS process, 3D printing, and post-processing methods.**

| | Scalability for scale-up production | Limitations | Compactness [piezoceramic ratio in green bodies]* | Capacity for forming a complex geometry | Piezoelectricity [$d_{33}$ pC/N] |
|---|---|---|---|---|---|
| 3D Printing: Slurry-based[19,20,22] | Low* | High viscosity requirement on the feedstock, Weak mechanical quality | Low [up to 35 vol%] | Yes | Medium [39–345] |
| Bulk solid-based[27-29] | Low* | High viscosity requirement on the molten feedstock, Weak mechanical quality | Low [up to 29 vol%] | Yes | Low [8.72] |
| Powder-based[23-25] | Low* | Not suitable for fabricating low-porosity components, Weak mechanical quality | Low [-] | Yes | Low [2.1] |
| Post processing: Thermal-based[31-34] | Medium* | Not suitable for non-laminated structures, limited applicability on various materials | High [-] | No | Medium [~390] |
| Prestressed[35,36] | Medium* | | High [-] | No | Medium [~390] |
| GDS process: | High* | Geometrical limits*, Not suitable for buried sintering | High [99 vol%] | Yes | High [~595] |

See Supplementary Note 3 for a detailed explanation of parameters with superscript *.
[-] denotes that this parameter has not been mentioned in the corresponding reference.

initial stage during the debinding process (Fig. 2a photographs I and II). After this process, the green compact tends to deform with rising temperature, resulting in a configuration with considerable curvature (Fig. 2a photographs III and IV). The curved configuration formed in the heating stage and early sintering stage determines the geometry of the final sintered body (Fig. 2a photographs V to X, Supplementary Fig. 1). The deformability of the sintering body is superior to that of ceramic sintered body[37], which is attractive for shape modeling.

To characterize the as-fabricated PZT ceramic, we used scanning electron microscopy (SEM) to show the well-densified ceramic without significant porosity (Fig. 2b and Supplementary Fig. 3a for the curved and flat specimen, respectively). X-ray diffraction experiments were performed on the sintered PZT ceramics and the result shows a pure tetragonal phase (Fig. 2c).

All elements (Pb, Zr, and Ti) in PZT are detected and distributed homogeneously in the cross section of the curved sample as verified by the EDS qualitative element mapping (Supplementary Fig. 2). Figure 2d shows the ferroelectric polarization versus electric field (P-E) loops of sintered PZT ceramics at room temperature (RT) and at 10 Hz. The remanent polarization ($P_r$) of the curved PZT ceramic is slightly higher than that of the flat sample. Both curved and flat PZT ceramics possess butterfly-like shape strain-electric field (S-E) curves (Supplementary Fig. 3b). The field-induced strains of the two specimens follow a similar trend and exhibit the maximum compressive strain of around 0.17%. Furthermore, we investigated the temperature dependence of the relative dielectric constant $\varepsilon_r$ of the two specimens (Supplementary Fig. 3c) and the corresponding Curie temperatures were identical (353.2 °C). To evaluate the mechanical

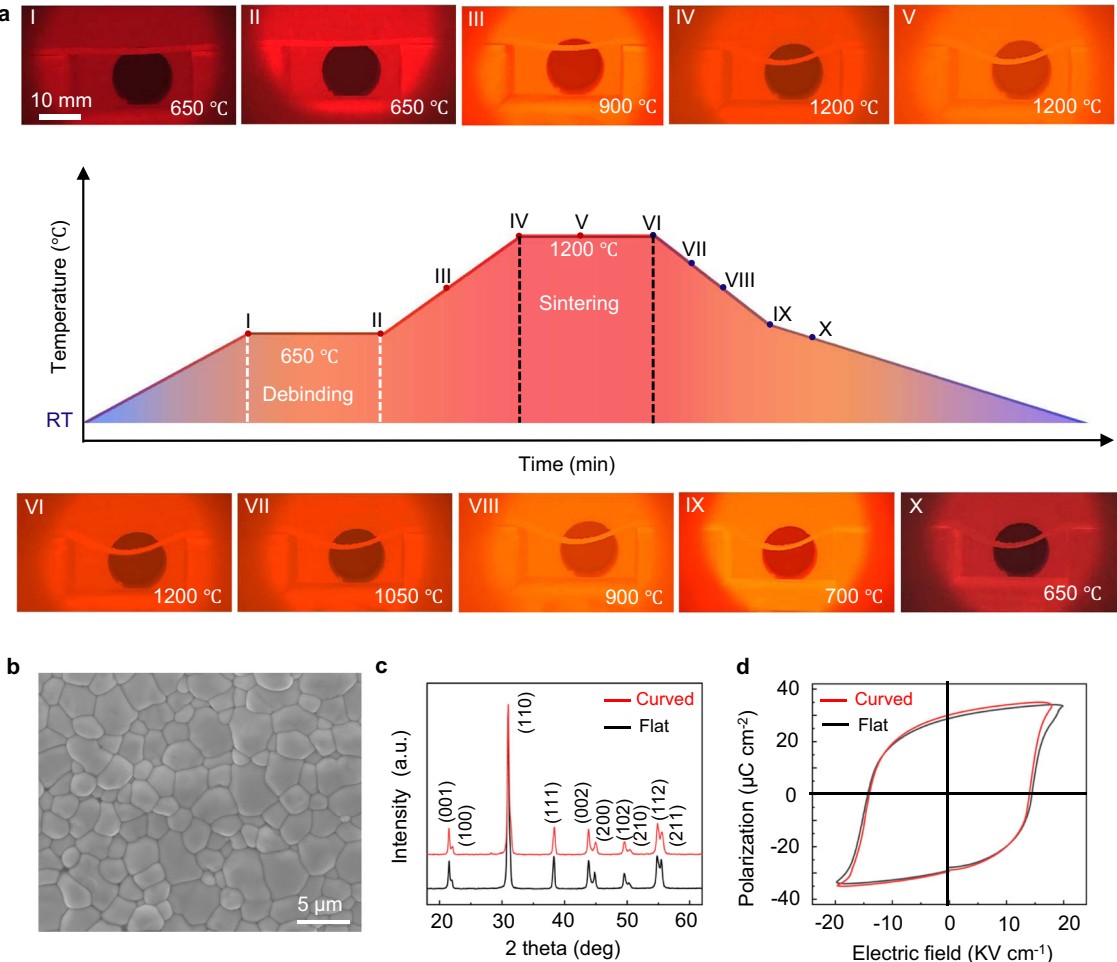

**Fig. 2 Fabricating PZT ceramic via the gravity-driven sintering (GDS) technique. a** Typical temperature profile of the GDS process. The photograph I to X demonstrate the formation process of curved geometry, where the curved configuration forms during III to IV and it maintains from IV to the end. RT represents room temperature. **b** SEM image of the curved surface of the obtained PZT ceramic. **c** The XRD patterns of the obtained curved and flat PZT ceramic. The two specimens exhibit an identical crystal structure (tetragonal phase). **d** Room temperature *P-E* hysteresis loops of the curved and the flat PZT ceramic measured at 10 Hz, indicating that the GDS-sintered and conventional-sintered PZT ceramics possess similar ferroelectric performance.

quality of the sintered ceramics, we perform tensile tests on the unpoled specimens cut from the curved and flat ceramics (see Methods, Supplementary Note 1, and Supplementary Fig. 4 for detailed experiment setup). The two ceramic specimens show close tensile force-displacement curves (see Supplementary Fig. 4d and Supplementary Note 1 for details) with a tensile strength of around 43 MPa for the flat sample and 36 MPa for the curved sample which is comparable to the values recorded in the previous works[47,48]. The curved and flat specimens also possess similar dielectric loss tanδ (around 0.020), piezoelectric constant $d_{33}$ (around 600 pC/N), and density (around 7609 kg/m³, see Supplementary Table 1 for detailed data). Following the aforementioned discussion and characterization, the GDS process enables us to fabricate curved PZT ceramic that exhibits identical crystal structure with comparable or better ferroelectric performance with respect to the conventional-sintered PZT ceramics.

**Performances and working principles of the GDS process.** To study the tuneability of the GDS process, we fabricated curved PZT ceramics under mechanical boundary conditions as schematically shown in Fig. 3a. Here, the effective length $l_e$ (Fig. 3a) works as the design variable. Meanwhile, we use dome height $h$ to characterize

the sintered specimens as the curvature varies along the length direction (see Methods for detailed explanation). The corresponding dome height $h$ of ceramics with different effective length $l_e$ are shown in Fig. 3b where the narrow error bar at each point indicates the good reproducibility of the GDS method. We found that the value of $h$ starts from around 0.74 mm at $l_e$ of 20 mm and it increases to 13.24 mm at $l_e$ of 42 mm. With proper boundary conditions, our GDS method exhibits good tuneability in a particular range of $l_e$, namely, the tuning region. This phenomenon indicates that to regulate the configuration of the GDS-fabricated products, two conditions must be satisfied: an appropriate $l_e$ should be designed, and the newly formed mechanical boundary should be stable enough to support the final product. It is worth mentioning that tuning $h$ by modifying $l_e$ in region C is easier and more stable than in regions B and D, which is evidenced by the slope evolution of the curve in the tuning region (Fig. 3b) and the scanning profiles of fabricated samples listed in Fig. 3c (see Supplementary Fig. 5 for the corresponding photographs).

To understand the differences in tuneability, we construct a simplified model in which gravitational force $F_g$ is balanced by the equivalent resistance force $F_r$ as schematically shown in Fig. 3d. In order to form a new equilibrium in configuration two (model with deep color) after breaking the initial equilibrium (shown as

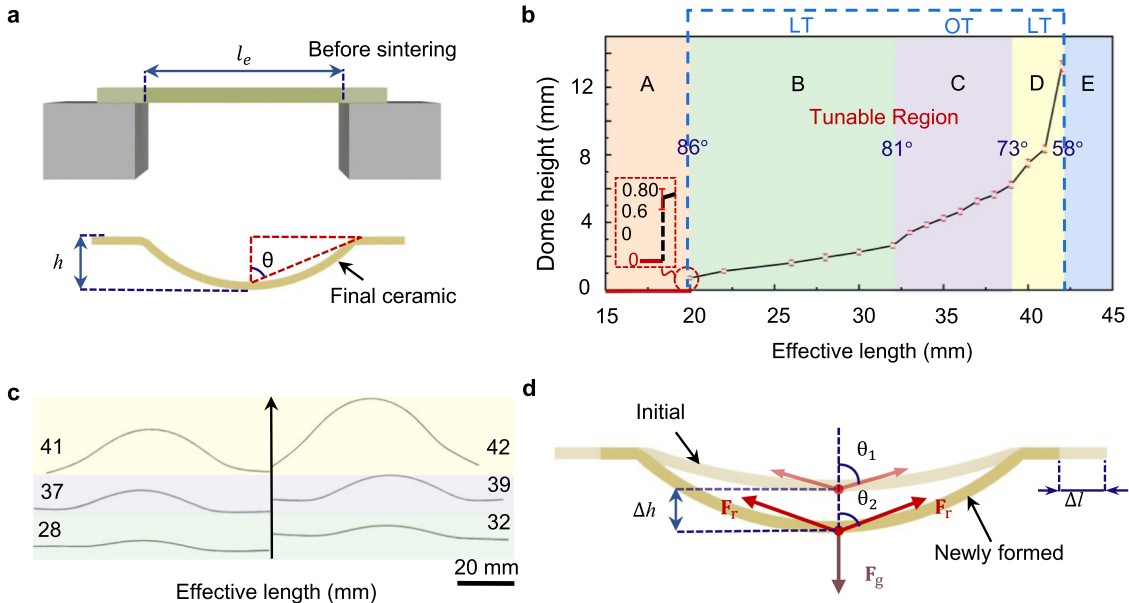

**Fig. 3 Reliability and tunability study of the GDS method. a** Schematic illustration of the two key variables: dome height $h$ and effective length $l_e$. **b** The relation of dome height $h$ with respect to effective length $l_e$. The blue dashed box highlights the tunable region which is defined into two segments, namely, the low tunability (LT) region and optimal tunability (OT) region. The narrow error bars demonstrate that the GDS process is reliable. **c** Scanning profiles of the surfaces of six specimens in regions B, C, and D, respectively. The numbers marked on the profiles denote the values of $l_e$. **d** Schematic of the simplified model, where the effect of viscosity behavior of the quasi-liquid specimen is simplified as two equivalent resistance forces: $F_r$ that balances the gravitational force $F_g$.

the light model), a sufficiently large resistant force is required that is mainly determined by the angle θ and the change of the effective length $\Delta l$. With a qualitative analysis on the equilibrium reestablishment of the newly formed configuration, a moderate $\Delta h / \Delta l$ is responsible for an optimal tuning region (region C). For instance, in region B, the large value of θ corresponds to a small vertical component of $F_r$, which means that a slight change of deformation $\Delta h$ usually requires a huge $\Delta l$ (see Supplementary Fig. 6 and Supplementary Note 2 for explanations of regions C and D).

To demonstrate the generality and customizability of the GDS process, we implement our method in several scenarios. The GDS process enables cosintering of multiple specimens simultaneously, which is friendly for scale-up production of curved piezoceramics. In practical piezoceramic fabrication, shaping can be a time-consuming process, especially when the final products possess diverse designed configurations. However, with the GDS sintering technique, 24 PZT ceramics with three different configurations can be cosintered using an 8 by 3 matrix setup (Fig. 4a schematic and photos of the sintered samples). Gravity-driven shaping at a high temperature can also generate deformation on other solid solution systems. To demonstrate this ability, we fabricate a BaTiO₃ (BT) curved ceramic (Fig. 4b). Comparing with PZT specimens (Fig.4a, column 3), the BT specimen (Fig. 4b, column 3) is less deformed after the GDS process although a large value of $l_e$ is adopted. This phenomenon is attributed to the small density of BT powders. We also sintered a large-curvature and a doubly-curved PZT ceramics by designing corresponding mechanical boundary conditions as shown in Fig. 4c (column 2 for schematics and column 3 for photographs). Additionally, a green compact can be placed on a designed mold to duplicate the geometry of the mold contact surface. To demonstrate this point, Fig. 4d shows the schematic and result for a PZT ceramic supported by a half alumina tube during the GDS process. In this scenario, the sintered ceramic shows an identical curvature with the inner surface of the supporting tube (Fig. 4d, column 3).

## Discussion

This work presents a method of fabricating geometrically complex piezoceramics with compact sintered bodies. The proposed GDS method exploits the softening effect during sintering for shape forming of piezoceramics. These GDS-sintered piezoceramics, with designable configurations and high compactness, eliminate any compromise between processibility and functionality of piezoceramics from 3D printing technology, thus suggesting applications for high-performance actuators, sensors, and energy harvesters. Considering that high-temperature induced viscous effect exists for various functional ceramics during sintering, we are convinced that the proposed methodology is not restricted to lead-based (PZT) and lead-free (BT) piezoceramic systems presented here. Beyond gravity, we expect to see many other driving forces for using thermal and force coupling effects to fabricate complex-shaped functional ceramics with compact bodies and without compromising functionalities.

## Methods

**Fabrication of the curved ceramic.** The polyvinyl alcohol binder (4% by weight of powders) was added to the purchased Pb(Zr$_{0.52}$Ti$_{0.48}$)O₃ (PZT) powders (Suzhou PANT Technology Co., Ltd.). Then, powders were uniaxially pressed to rectangle green compacts with dimensions of 80 mm × 30 mm × 1.15 mm at 20 MPa in a steel mold. After the binder burned out at 650 ºC, the green compacts were sintered at 1200 ºC for 2 h. BaTiO₃ ceramics were prepared by adding polyvinyl alcohol as a binder to purchased BaTiO₃ powders (Suzhou PANT Technology Co., Ltd.). The powders were pressed into rectangle green compacts with dimensions of 80 mm × 30 mm × 1.4 mm at 20 MPa and sintered at 1300 ºC for 2 h after debinding at 600 ºC for 2 h.

**Material characterizations.** The phase structures of the curved and flat PZT ceramics were measured by X-ray diffraction (Rigaku SmartLab) with a scan speed of 4° per min. The microstructures of the surfaces of PZT ceramics were observed with scanning electron microscopy (SEM; FEI Quanta 450). Elemental mapping (EDS) was detected by an energy dispersive spectrometer (Oxford Instruments, INCA Energy 200). The sintered ceramics were carefully cut and polished in the form of a slice with a thickness of 1 mm for ferroelectric measurements and 0.4 mm for electro-strain behavior measurement. Silver electrodes were deposited onto the surface of cut specimens with magnetron sputtering (Q150TS). The ferroelectric

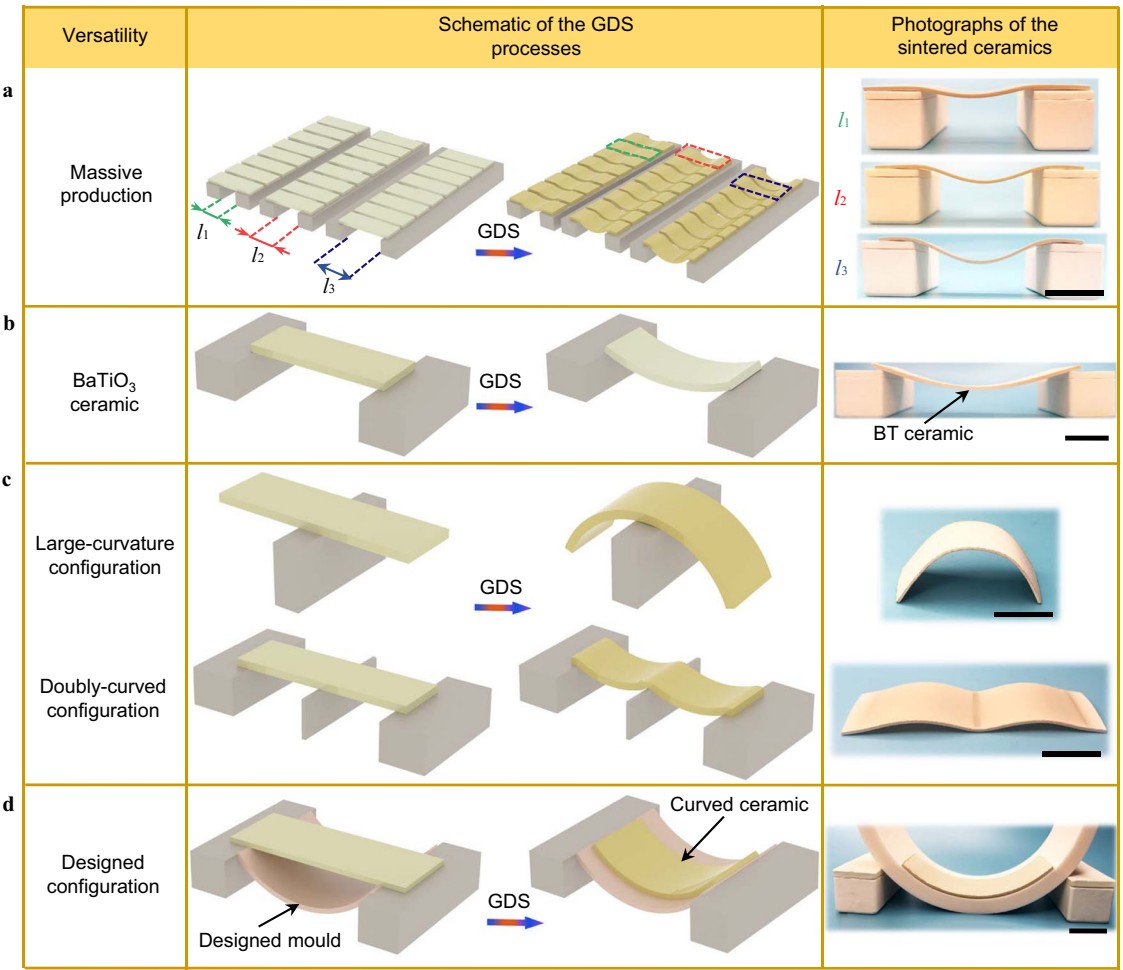

**Fig. 4 A general fabrication framework for materials and configurations enabled by the GDS sintering technique.** All scale bars are 15 mm in column 3. **a** Schematic of an 8 by 3 matrix for cosintering 24 curved ceramics with the GDS technique (column 2) and photographs of the corresponding sintered specimens (column 3). **b** Schematic of implementing the GDS method on the BaTiO$_3$ (BT) ceramic (column 2) and photographs of the obtained curved BT ceramic (column 3). **c** Schematics of fabricating PZT ceramics with diverse configurations by designing initial mechanical boundaries (column 2). The fabricated ceramics exhibit the same geometries with the corresponding designs demonstrated in column 2. **d** Schematic of the GDS method for fabricating PZT ceramics with a designed curved surface, in which the sintered ceramic possesses the same curve surface with the inner surface of the alumina support.

hysteresis loop was tested with a ferroelectric analyzer (PK-CPE 1701, PolyK Technologies, USA). The strain-electric field curves were measured with a ferro-electric measurement system (RTI-LC II, Radiant Technologies Inc, USA) and a laser measurement system equipped with a strain gauge amplifier (MTI-2100, Hottinger Baldwin Messtechnik GmbH, Darmstadt, Germany). The tensile force-displacement curves were measured with a tensile tester system (TY8000-A, Tian Yuan Test Instrument). The density was measured by the Archimedes method. The quasi-static piezoelectric constants were measured with a quasi-static piezoelectric meter (YE2730A d33 meter). Temperature dependence of the relative dielectric constant and corresponding dielectric loss tanδ were detected by a high-temperature dielectric property test system (DPTS-AT-600, Wuhan Yanhe Technology Co., Ltd).

**Dome height and surface profiles measurements**. For each preset $l_e$, dome heights of six specimens fabricated three times were measured. The dome height $h$ was measured with a micrometer. The surface profiles shown in Fig. 3c were obtained by scanning projections of the curved specimens along the width direction.

**Tensile tests of unpoled ceramics**. The unpoled ceramics are first cut and polished in the form of a slice with the dimension of 25 mm × 5 mm × 0.7 mm. To avoid the fracture at the gripping area, we design four tabs as shown in Supple-mentary Fig. 4. The glass-fiber tabs are bonded to both sides of the slice specimens and then clamped by the tensile machine for measuring the force-displacement curves. The high measured displacement of each sample originates from the addition of the deformation of the tabs.

## Data availability

The authors declare that data supporting the findings of this study are available from the corresponding author upon reasonable request.

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

## Acknowledgements

The work described in this paper was supported by General Research Grant (Project Nos. CityU 11212021) and Early Career Scheme (Project No. CityU 21210619) from the Research Grants Council of the Hong Kong Special Administrative Region, and Shenzhen Fundamental Research Program (No. JCYJ20200109143206663).

## Author contributions

Z.Y. and Y.S. initialed the project and conceived the experiments. Y.S. fabricated the samples, performed the materials characterization, and analysed data with assistance from S.L., B.W., Y.H. and C.Z. C.W.L. provided theoretical support for the force analysis. All authors commented on the manuscript.

## Competing interests

The authors declare no competing interests.
