## [Peer Review File · Nature Communications]

Title: A gravity-driven sintering method to fabricate geometrically complex compact piezoceramicsREVIEWER COMMENTS

Reviewer #1 (Remarks to the Author):

In this work, Shan et al. proposed a gravity-driven sintering (GDS) method to fabricate geometrical piezoceramics, and some results can be realized. However, the present results in this work cannot be recommended to the publication of Nature Communications. I addressed my concerns in the following.

1. Did the authors carry out the mechanical measurements to compare the mechanical quality of their geometrical ceramics and others (such as fabricated by CET-based, mechanically pre-stressed, and 3D printing methods)?
2. On page 3, the authors highlighted the ability of GDS process in Table 1. However, they did not list the mechanical properties that are the key points as they addressed in the Introduction.
3. In Figure 2B, the authors are suggested to add the SEM image for flat PZT ceramic for comparison.
4. In Figure 2D, the authors only measured P-E loops for curved and flat PZT ceramics. The authors are suggested to measure the electric field-induced strain (S-E) curves to some verify whether the deformation by applying electric fields affects (or even breaks) the geometry because of the break and reestablishment of force equilibrium. This point is very important for practical applications.
5. In Figure 3, dome height (h) starts from around 0.74 mm at an effective length (le) of 20 mm. Therefore, the curved PZT ceramics by GDS process must contain enough raw materials and can be only used for very limited areas (e.g., requiring piezoceramics with a large size). However, as we know, miniaturization is the tendency for future electronic devices. Therefore, how do the authors think about this disadvantage?
6. In Figure 4, the authors demonstrated the generality and customizability of the GDS process. They tried the GDS process on preparing BaTiO₃ ceramics that are not volatile. It seems like that the GDS process cannot be carried out for the buried sintering. Therefore, the GDS process does not apply to other piezoceramics containing volatile elements, such as alkalis (K, Na, Li) and bismuth (Bi).

Reviewer #2 (Remarks to the Author):

In this work, the authors report a new method of gravity-driven sintering for fabricating geometrically complex piezoceramics. The fabricated curved PZT piezoceramics show material properties and ferroelectric performance comparable to those of the conventional-sintered ones. This work provides a facile route to fabricate high-performance curved piezoceramics. I recommend its publication in Nature Communications.

1. Since there are different kinds of PZT (Science 2019, 363, 1206-1210), the authors need to mention the compositions of the PZT in this work.
2. From Table 1, the gravity-driven sintering method fabricated curved PZT piezoceramics present an enhanced piezoelectricity compared to those fabricated by other methods. The authors are suggested to discuss the reason in detail.

3. For the geometry of gravity-driven sintering method fabricated piezoceramics, please discuss its controllability and reproducibility.

Reviewer #3 (Remarks to the Author):

Nice and clear paper on using creep on materials during sintering to form shape.

I have relatively minor comments.

Some formatting issues/symbols missing in pdf e.g. oC (including supplemental)

Some typo Mpa - need to be MPa (many instances)

Methods:

“purchased PZT” please state company and type, why selected? This looks like a soft PZT eg. PZT-5H. Is there any reason this would be better suited? What about harder PZT materials?

Fig 3 a -The dome should be the other way around to be more clearly linked to how it is formed? Or is this the mould? If it is the mould make a different colour to PZT/final ceramic. Either way, this should be clear.

Avoid “Massive production” - use ‘scale-up’ or another term.

For the table - the GDS process limitations are ‘NONE’ - there are clearly some limitations to the methods and these should be clearly stated - for example, it is more suited to curved shapes - for example, it is not possible to produce rings or squares shapes for transducers. Please include geometrical limits here.

Overall nice paper.

CR Bowen

Reviewer Comments (and change made in accordance)

Reviewer #1

In this work, Shan et al. proposed a gravity-driven sintering (GDS) method to fabricate geometrical piezoceramics, and some results can be realized. However, the present results in this work cannot be recommended to the publication of Nature Communications. I addressed my concerns in the following.

Response: We appreciate that the referee proposes these constructive and insightful suggestions for improving the quality of this article. Our work presents a paradigm for fabricating geometrically complex piezoceramics via exploiting the high-temperature viscous behavior of the ceramic material during sintering. The proposed GDS method bypasses processing fragile ceramic green bodies and sintered samples, thereby creating the freedom to facilitate design geometrical functional ceramics. The GDS method is applicable for forming plentiful geometries for various functional ceramics, although some issues exist in present work. We envision that the design principle of the GDS methodology will inspire researchers to explore potential processing methods for materials under multi-physical coupling effect. Following the referee's comments and suggestions, we have revised our manuscript with the most seriousness (the revised portions are marked in Red Color in the manuscript) and the detailed responses are listed below.

Questions and comments:

1. Did the authors carry out the mechanical measurements to compare the mechanical quality of their geometrical ceramics and others (such as fabricated by CET-based, mechanically pre-stressed, and 3D printing methods)?

Response: We thank the referee for this constructive comment. We agree that the mechanical performances are important for practical applications of functional ceramics. The mechanical qualities like tensile strength and stiffness of one structure are strongly related to its geometrical design and the properties of constituent materials. Therefore, the material properties (porosity in particular) are main factors to the mechanical performances of geometrical ceramics in identical geometry. A key to understand the mechanical properties of ceramic materials is to determine the tensile strength via tensile tests. To avoid

fracture at the gripping area, the tensile tests are usually performed on testing specimens in dog-bone shapes. However, few works report the tensile tests of piezoceramics due to the difficulties in fabricating required testing specimens. Here, we measure the tensile strength (Supplementary Table S1) to quantitatively demonstrate the mechanical quality of the fabricated PZT ceramics. The sintered ceramic samples are carefully cut and polished in the form of slice with thickness of 0.7 mm, and then the tapered tabs are bonded to both sides of the unpoled slice specimens to prevent fracture at the gripping area (please check more detail in the reference [47][48] listed in the revised manuscript) as schematically shown in Fig S4 in the revised Supplementary Materials. We also list the figure, the corresponding demonstration (Results, Methods in the revised manuscript and Supplementary Text 3 in the revised Supplementary) and reference here for the convenience of the reviewers.

Supplementary Figure S4 | Tensile test for evaluating the mechanical quality of fabricated ceramics (a) Schematic and (b) key parameters of the testing specimen and the tapered tabs. (c) Photographs of the testing system and clamped sample. (d) Tensile force-displacement curves of the curved and flat PZT ceramics measured with tensile loading (displacement) of 0.5 mm/min, where the corresponding fracture occurs on the middle of the samples (inset).

Supplementary Text 3 | Explanation of the mechanical measurement (on page 2 in the revised supplementary)

The mechanical properties of ceramics are important for practical applications. Tensile strength is usually used to evaluate the mechanical quality of ceramic specimens. Normally, to avoid fracture at the gripping area, the testing specimens are designed and fabricated in the dog-bone shape^{[47],[48]}. However, difficulties in fabricating the dog-bone piezoceramic specimens hinder conducting tensile tests on piezoceramics. To solve this problem, we designed four tapered tabs to clamp the PZT slices as schematically shown in Supplementary Figure S4 A and B. The PZT slices are bonded on the glass-fiber tabs, and then the testing specimens are clamped by the tensile machine (Supplementary Figure S4 C). It should be mentioned here that the clamping strategy used in this work can effectively prevent the testing PZT slices from fracture during the initial clamping state as the clamping forces are directly applied on the four tabs. The force-displacement curves are shown in Supplementary Figure S4 D, where the insets exhibit the fracture of the PZT slices. Although this special clamping method causes high measured deformation, the tensile strengths of the testing specimens are comparable with the recorded values (around 40~50 MPa) as the fracture caused by the tensile force occurs at the middle of the PZT slice.

Results (at lines 153-159 on page 4 in the revised manuscript)

To evaluate the mechanical quality of the sintered ceramics, we perform tensile tests on the unpoled specimens cut from the curved and flat ceramics (see Methods and Supplementary Figure 4 for detailed experiment setup). The two ceramic specimens show close tensile force-displacement curves (see Methods, Supplementary Text3 and Supplementary Fig. 4 for detailed experiment setup) with tensile strength of around 43 MPa for flat sample and 36 MPa for the curved sample which is comparable to the values recorded in previous works^{[47],[48]}.

Methods (on page 5 and 6 in the revised manuscript)

The tensile force-displacement curves were measured with a tensile tester system (TY8000-A, Tian Yuan Test Instrument). (at lines 243-244 on page 5 in the Material characterizations part)

Tensile tests of unpoled ceramics. The unpoled ceramics are first cut and polished in the form of the slice with the dimension of 25 mm × 5 mm × 0.7 mm. To avoid the fracture at the gripping area, we design four tabs as shown in supplementary Figure 4. The glass-

fiber tabs are bonded to both side of the slice specimens and then clamped by the tensile machine for measuring the force-displacement curves. The high measured displacement of each sample originates from the addition of the deformation of the tabs. (at lines 253-258 on page 6)

In the revised manuscript, we also show the Surface morphology, Phase structure, Ferroelectricity, Elemental distribution, Electro-strain behavior, Thermal stability and Dielectric performance of the fabricated curved PZT ceramics in Fig. 2B-D, Fig. S2, Fig. S3 and Table. S1, respectively. Based on these characterizations, we demonstrated that the GDS method enables us to fabricate geometrical PZT ceramic that exhibits identical material properties with respect to the conventional-sintered flat PZT ceramics (**Results** on page 3 and page 4 in the revised manuscript). This means that the GDS method has neglectable effects on the nature of the processing piezoceramic material including the mechanical and functional properties compared with these conventional-sintered piezoceramic counterparts. Here, instead of directly measuring the mechanical properties, we qualitatively compare the mechanical qualities of products fabricated by CET-based, Mechanically pre-stressed and 3D printing with our GDS method below:

1. CET-based and Mechanically pre-stressed methods directly process PZT ceramic sintered bodies. In these two kinds of fabrication methods, conventional-sintered PZT ceramics are adhered on metallic substrates to form thermal/mechanical stress mismatch between ceramic and metal layers when under heating/tensile treatment. The final products of the two methods are laminates comprised of slightly curved piezoceramics and metal substrates. The mechanical performances of the curved PZT layer are determined by the initial PZT specimens obtained via conventional-sintered method as the post process (heating/tensile) is elaborately carried out. Unlike CET-based and mechanically pre-stressed methods, our GDS method works on the ceramic sintering bodies and generate curved piezoceramics in compact body finally. We have demonstrated that the GDS-fabricated PZT ceramics possess the same material properties as conventional-sintered PZT specimens. So, the CET-based, Mechanically pre-stressed and the GDS methods have negligible effects on the mechanical properties of fabricated curved piezoceramics. Besides, the mechanical performances of the curved piezoceramics fabricated via CET-based and Mechanically pre-stressed methods have never been discussed because it is difficult to separate the fragile piezoceramic layers from metallic substrates.
2. 3D-printing techniques are powerful for forming geometrically complex ceramic green

bodies. To print geometrically stable and accurate piezoceramic green bodies, high volume of organic binders (normally over 40 vol%) is required in feedstocks. These 3D-printed piezoceramic green bodies become porous sintered bodies after sintering process, thereby compromising the mechanical and piezoelectric performances. Besides, the layer-by-layer fabrication process also causes poor mechanical performances of printed specimens as the weak bonding force between layers decrease the mechanical strength of the specimens. The detailed discussion on the mechanical properties of 3D-printed piezoceramics have been well reviewed by Cheng Chen et.al. (Reference [4] in the revised manuscript) and they concluded that weak mechanical properties of fabricated piezoceramics are the main limitation for most 3D-printing ceramics techniques. Here, we list the comparison Table shown in Chen’s work for the convenience of referees.

Table 2. Comparison of additive manufacturing techniques for piezoelectric devices^[65,79,109].

Process	AM method	Available materials	Advantage	Limitation	Applications
Polymerization	SLA	Photocurable resins, photopolymers, ceramic, metal or polymer particles in the resin	Simple operation method, high resolution, high surface precision, high efficiency	Not suitable for multimaterial mixing, high slurry requirements, not suitable for the preparation of large devices	Rapid tooling patterns, snap fits, very detailed parts, presentation models, high heat applications
Extrusion	EFF	Polymer, sol-gel	Simple operation method, no need to demould, Not easily deformed, no mask required	Low density, low surface finish, unable to prepare microdevices	Rapid mass production of large devices, making of simple food applications
Extrusion	FDM	Thermoplastic, polymers, polymer-based nanocomposites	User-friendly, low cost, large size capabilities, wide range of applications	Weak mechanical properties, high viscosity of the molten materials, low surface quality	Small detailed parts, presentation models, patient and food applications, high heat applications
Extrusion	DIW	Thermoplastics, sol-gel, metal, ceramics	Affordable cost, easy operation, large material diversity, and no mask requirement	Weak mechanical properties, high ink requirements	Very detailed parts, rapid tooling patterns, jewelry and fine items
Powder-based	SLS	Thermoplastics, metal	Bulk objects from various powdered materials, no mould required, low cost	Material surface fineness, low surface quality, low resolution	Less detailed parts, parts with snap-fits and living hinges, high heat applications
Powder-based	NFES	Ceramics, polymer, composite	Increase piezoelectric output, simple, economical and versatile	Weak mechanical properties, complicated subsequent operations	Making free-form piezoelectric devices, production of bulk materials
Powder-based	EPAM	Piezoelectric polymer material, piezoelectric ceramic material	Good uniformity in production and simple fabrication step, low cost, single processing	Weak mechanical properties, high ink requirements	Making free-form piezoelectric devices, suitable for sensing, actuation with simplicity
Solvent-based	SEA-3DP	Polymer, composite	Scalable, rapid, versatile, high efficiency, simplicity, versatility, and controllability	High material requirements, not good for high volume printing, and print head is less durable	Suitable for small, simple structures, making free-form piezoelectric devices

Following the aforementioned demonstration, the geometrical piezoceramics fabricated by the GDS, CET-based and Mechanically pre-stressed methods possess comparable mechanical properties with conventional-sintered piezoceramics. However, the piezoceramics fabricated via 3D-printing methods exhibit compromised mechanical performances.

Reference

4. Chen, C. et al. Additive manufacturing of piezoelectric materials. *Adv. Funct. Mater.* 30, 1–29 (2020).

47. Ceramics, P. et al. Tensile Stress-Strain Behavior of Piezoelectric Ceramics. *Jpn. J. Appl. Phys.* 32, 4233–4236 (1993).

48. Fett, B. T. & Munz, D. Deformation of PZT Under Tension , Compression , Bending , and Torsion Loading. *Adv. Eng. Mater.* 5, 718–722 (2003).

2. On page 3, the authors highlighted the ability of GDS process in Table 1. However, they did not list the mechanical properties that are the key points as they addressed in the Introduction.

Response: We thank the reviewer for this insightful comment. According to the reviewer's suggestion, we have added description of mechanical qualities of the products in Table 1. Please check Table 1 on page 12 in the revised manuscript. We also list the revised Table 1 here for the convenience of reviewer.

Table 1 | Comparison of processing methods for fabricating compact piezoceramics with complex geometries by the GDS process, 3D printing and post-processing methods.

	Scalability for scale-up production	Limitations	Compactness [piezoceramic ratio in green bodies]*	Capacity for forming complex geometry	Piezoelectricity [d_{33} pC/N]
3D Printing:					
Slurry-based ^{20,21,23}	Low*	High viscosity requirement on the feedstock, Weak mechanical quality	Low [up to 35 vol%]	Yes	Medium [39-345]
Bulk solid-based ^{28,29,30}	Low*	High viscosity requirement on the molten feedstock, Weak mechanical quality	Low [up to 29 vol%]	Yes	Low [8.72]
Powder-based ^{24,25, 26}	Low*	Not suitable for fabricating low-porosity components, Weak mechanical quality	Low [-]	Yes	Low [2.1]
Post processing:					
Thermal-based ^{31,32,33,34}	Medium*	Not suitable for non-laminate structures, limited applicability on various materials	High [-]	No	Medium [~390]
Pre-stressed ^{35,36}	Medium*		High [-]	No	Medium [~390]
GDS process:	High*	Geometrical limits*, Not suitable for buried sintering	High [99 vol%]	Yes	High [~595]

3. In Figure 2B, the authors are suggested to add the SEM image for flat PZT ceramic for comparison.

Response: We highly appreciate the reviewer's suggestion. We have added the scanning electron microscopy (SEM) of the flat PZT ceramic in Supplementary Figure S3A in the revised Supplementary Materials. We have added the demonstration of this new figure at lines 142-143 on page 3 in the revised manuscript. We also list the new SEM image and the revised sentence here for convenient.

To characterize the as-fabricated PZT ceramic, we used scanning electron microscopy (SEM) to show the well-densified ceramics without significant porosity (Fig. 2B for the curved specimen and Fig. S3A for the flat specimen).

Supplementary Figure S3 | **Material properties characterization of fabricating ceramics** (a) SEM image of the flat PZT ceramic. (b) Room-temperature S-E hysteresis loops of the curved and the flat PZT ceramic measured at 1 Hz. (c) Temperature dependence of relative dielectric constant ϵ_r of curved and flat PZT ceramics measured at frequency 1 kHz.

4. In Figure 2D, the authors only measured P-E loops for curved and flat PZT ceramics. The authors are suggested to measure the electric field-induced strain (S-E) curves to some verify whether the deformation by applying electric fields affects (or even breaks) the geometry because of the break and reestablishment of force equilibrium. This point is very important for practical applications

Response: We thank for the referee's constructive comments. Following the referee's suggestion, we have measured the strain-electric field (S-E) curves of the curved and flat specimens. The sintered ceramics were carefully cut and polished in the form of slice with thickness of 0.4 mm for electro-strain behavior measurement. The strain-electric field curve is measured with a ferroelectric measurement system (RTI-LC II, Radiant Technologies Inc, USA) and a laser measurement system equipped with a strain gauge amplifier (MTI-2100,

Hottinger Baldwin Messtechnik GmbH, Darmstadt, Germany). The S - E curves are shown in Figure S3B in the revised Supplementary Materials. We have added the demonstration of Figure S3B at lines 149-151 on page 4 in the revised manuscript. For the convenience of reviews, we also list the S - E curve and the corresponding demonstration below.

Both curved and flat PZT ceramics possess butterfly-like shape strain-electric field (S - E) curves (Supplementary Fig. 3B). The field-induced strains of two specimens follow the similar trend and exhibit the maximum compressive strain of around 0.17%.

Supplementary Figure S3 | **Material properties characterization of fabricating ceramics** (a) SEM image of the flat PZT ceramic. (b) Room-temperature S - E hysteresis loops of the curved and the flat PZT ceramic measured at 1 Hz. (c) Temperature dependence of relative dielectric constant ϵ_r of curved and flat PZT ceramics measured at frequency 1 kHz.

- In Figure 3, dome height (h) starts from around 0.74 mm at an effective length (l_e) of 20 mm. Therefore, the curved PZT ceramics by GDS process must contain enough raw materials and can be only used for very limited areas (e.g., requiring piezoceramics with a large size). However, as we know, miniaturization is the tendency for future electronic devices. Therefore, how do the authors think about this disadvantage?

Response: We sincerely thank the referee for this insightful comment. In this manuscript, we demonstrate that using creep on piezoceramic materials during sintering for forming geometries is feasible and superior than existing methods in terms of fabricating geometrical piezoceramics without compromising material properties. To verify the feasibility of the GDS method, we performed the GDS method on PZT green compacts with dimension of 80 mm \times 30 mm \times 1.15 mm. As schematically illustrate in Figure 1B, the green compact tends to deform when the shear force cannot balance the gravitational force of the effective segment. The sintering specimen stop deforming when the gravitational force is rebalanced by the shear force (internal shear force and shear force).

Therefore, the key factor to the final geometry of the specimen is the force equilibrium in the sintering system (supporting blocks and the ceramic specimen). To better understand this deformation process and force equilibrium, considering the supporting blocks and the PZT green compact as a simply supported beam at two ends. It is easy to understand that the gravitational force is directly linked to the volume of the effective segment. However, the shear forces are related to the geometry of the cross section (thickness and width of the specimen) and the volume of the supported segment. As shown in Figure 3A, the green compact is supported by two alumina blocks and the length of the supported segment (l_s) can be written as:

$$l_s = (l - l_e)/2$$

where l denotes the length of the green compact. In this manuscript, the value of l is given to 80 mm in all specimens shown in Figure 3. For the case ($l_e = 20$ mm), the small dome height originates from the low value of the ratio of l_e to l_s . So, the large size is not highly required for using the GDS method as the force equilibrium is determined by the designed mechanical boundary conditions (the ratio of l_e to l_s in cases presented in this manuscript). We agree with the reviewer that miniaturization is the tendency for future electronic devices. Our GDS method is available for shaping small-size piezoceramics as the ratio of l_e to l_s is dimension independent. We also applied the GDS method on PZT green compact in dimension of 30 mm × 10 mm × 0.6 mm and successfully fabricate curved PZT ceramic (see Figure 4C in the revised manuscript). We highlighted in the manuscript that the final geometry of the PZT ceramic is determined by the initial mechanical boundary condition. Besides, we are working on constructing micro robots with our GDS-fabricated functional ceramics recently.

6. In Figure 4, the authors demonstrated the generality and customizability of the GDS process. They tried the GDS process on preparing BaTiO₃ ceramics that are not volatile. It seems like that the GDS process cannot be carried out for the buried sintering. Therefore, the GDS process does not apply to other piezoceramics containing volatile elements, such as alkalis (K, Na, Li) and bismuth (Bi).

Response: We thank the referee for this constructive comment. In this manuscript, we implement the GDS method on PZT and BT solid solution system to illuminate the generality of this method. We admit that the GDS method is not suitable for buried sintering as sufficient domains are required for sintering specimens to deform. In the PZT solid solution system, Pb element is volatile during sintering, thereby causing poor piezoelectricity (low d_{33}). To solve this problem, excessive amounts of Pb element is

usually required to create a lead atmosphere in the sealed sintering system (alumina crucible). Following this method, we putted several PZT blocks around the constructed GDS system in the sealed alumina crucible for forming Pb atmosphere. We also tried to fabricate geometrical PZT ceramic via GDS method without putting PZT blocks in the sealed crucible and the as-fabricated PZT ceramic exhibits inferior piezoelectric coefficient constant d_{33} of around 310 pC/N than that of specimens (around 595 pC/N) presented in the manuscript. Overall, the GDS method is practicable for piezoceramics containing volatile elements as the volatile element atmosphere can be created in the sealed system.

Reviewer #2

In this work, the authors report a new method of gravity-driven sintering for fabricating geometrically complex piezoceramics. The fabricated curved PZT piezoceramics show material properties and ferroelectric performance comparable to those of the conventional-sintered ones. This work provides a facile route to fabricate high-performance curved piezoceramics. I recommend its publication in Nature Communications.

Response: We are pleased that the referee acknowledges the novelty of our GDS method for fabricating curved piezoceramics. Thank the referee for recommendation of the publication on Nature Communications. Following the referee's comments and suggestions, we have revised our manuscript with the most seriousness (the revised portions are marked in Red Color in the manuscript) and the detailed responses are listed below.

Questions and comments:

1. Since there are different kinds of PZT (Science 2019, 363, 1206-1210), the authors need to mention the compositions of the PZT in this work.

Response: We thank the referee for this comment. In this work, we purchase the $\text{Pb}(\text{Zr}_{0.52}\text{Ti}_{0.48})\text{O}_3$ (PZT) powders from Suzhou PANT Technology Co., Ltd. We have mentioned the compositions of the used PZT in the Methods part in the revised manuscript.

2. From Table 1, the gravity-driven sintering method fabricated curved PZT piezoceramics present an enhanced piezoelectricity compared to those fabricated by other methods. The authors are suggested to discuss the reason in detail.

Response: We thank the referee for this insightful suggestion. We have discussed the reason why the GDS-fabricated curved PZT piezoceramics exhibit superior piezoelectricity than those fabricated by other methods on page 2 in the revised Supplementary Materials. We also list these sentences here for the convenience of the referee.

Piezoelectricity [d_{33} pC/N]

Piezoelectricity is the key property of piezoceramics, which denotes the ability to achieve

the electromechanical coupling. We use the piezoelectric constant d_{33} to characterize the piezoelectricity of the piezoceramics discussed in Table 1. The post processing methods directly process compact piezoceramic sintered bodies, therefore, the value of d_{33} is mainly determined by the piezoelectricity of the used piezoceramic layer. The differences in d_{33} between the GDS-fabricated and post-processing curved ceramics originate from the material composition of the purchased PZT material. However, for 3D printing methods, low volume of piezoceramic constituents in the printed green bodies predestine the as-sintered piezoceramics in porous bodies and thus compromised piezoelectricity (small value of d_{33}) compared with the other two kinds of methods.

3. For the geometry of gravity-driven sintering method fabricated piezoceramics, please discuss its controllability and reproducibility.

Response: We thank the referee very much for the valuable advice. In Figure 3, we studied the tuneability of the GDS method. We obtained the values of dome height h and effective length l_e from six groups of specimens fabricated in three different times (mentioned in the Methods part). The h versus l_e curve is shown in Figure 3B where the highly narrow error bar at each point indicates the good reproducibility of the GDS method. Additionally, in each tunable region (region B, C and D in Figure 3B), the value of h goes linearly with increasing l_e . This means that for the initial mechanical boundary condition in Figure 3A, the GDS method enables users to design curved piezoceramics in desired dome height via tuning the value of l_e . We have added corresponding description in lines 169-170 on page 4 in the revised manuscript. Besides, we also qualitatively analyzed the tuneability of the GDS method via constructing a simplified model in Figure 3D.

Reviewer #3

Nice and clear paper on using creep on materials during sintering to form shape. I have relatively minor comments.

Response: We thank the referee very much for the constructive comments and all suggestions upgrade the quality of our article. We are pleased that the referee acknowledges the novelty of our work on fabricating curved piezoceramics by exploiting the high-temperature creep effect on material. Following the referee's comments and suggestions, we have carefully revised our manuscript (the revised portions are marked as in **Red Color** in the revised manuscript). Please find our detailed response below.

Questions and comments:

1. Some formatting issues/symbols missing in pdf e.g. oC (including supplemental). Some typo Mpa - need to be MPa (many instances)

Response: We sincerely thank for the referee's careful examination and these mistakes have been checked and revised in the revised manuscript.

2. Methods: "purchased PZT" please state company and type, why selected? This looks like a soft PZT eg. PZT-5H. Is there any reason this would be better suited? What about harder PZT materials?

Response: We thank the referee for these insightful comments. The PZT powders are PZT-5H purchased from Suzhou PANT Technology Co., Ltd. We have added the detail of the used powders in the Methods part in the revised manuscript. There are no particular reasons for using this kind of PZT powders. The GDS method is available for other kinds of PZT powders (harder and softer) as the high-temperature viscous effect exists for various ceramic materials. As shown in Figure 4B, the differences in the properties of the used piezoceramic (BT ceramic) influence the deformation process during sintering, thereby causing different curved geometries with PZT specimens using identical mechanical boundary condition. The comparison between PZT and BT curved specimens illustrate that the shaping principle of our GDS method works in various piezoceramic materials, although the corresponding tuning patterns vary from material to material.

3. Fig 3 a -The dome should be the other way around to be more clearly linked to how it is formed? Or is this the mould? If it is the mould make a different colour to PZT/final ceramic. Either way, this should be clear.

Response: We thank the referee for these constructive comments and suggestions. In Figure 3A, we illustrate the two key variables namely dome height h (in sintered specimens) and effective length l_e (in green compacts). The formation process of the dome is schematically shown in Figure 1B and in-situ recorded in Figure 2A. The dome shown in Figure 3A is the final ceramic, and we have added labels in the revised Figure 3A to make it clear.

Fig.3 Reliability and tunability study of the GDS method. (A) Schematic illustration of the two key variables: dome height h and effective length l_e . (B) The relation of dome height h with respect to effective length l_e . The blue dashed box highlights the tunable region which is defined into two segments, namely, Low Tunability (LT) region and Optimal Tunability (OT) region. The narrow error bars demonstrate that the GDS process is reliable. (C) Scanning profiles of the surfaces of six specimens in region B, C and D, respectively. The numbers marked on the profiles denote the values of l_e . (D) Schematic of the simplified model, where the effect of viscosity behavior of the quasi-liquid specimen is simplified as two equivalent resistance forces: F_r that balances the gravitational force F_g .

4. Avoid “Massive production” - use ‘scale-up’ or another term.

Response: We sincerely thank for the referee’s careful examination. We have revised the words ‘Massive production’ to ‘Scale-up production’.

5. For the table - the GDS process limitations are 'NONE' - there are clearly some limitations to the methods and these should be clearly stated - for example, it is more suited to curved shapes - for example, it is not possible to produce rings or squares shapes for transducers. Please include geometrical limits here.

Response: We thank the referee for this constructive comment. The GDS method shows its superiority in producing geometrical piezoceramics with high compactness. Although there are some geometrical limits on our GDS method as kindly mentioned by the referee. We have revised Table 1 and demonstrated the geometrical limitations of the GDS method in the revised Supplementary Materials. In addition, the GDS method is not suitable for ceramics that highly rely on buried sintering because sufficient domains are required for sintering specimens to deform. We also list Table 1 and the corresponding sentence here for the convenience of the reviewer.

Geometrical limits

The GDS method are not suitable for fabricating piezoceramics in over than half-round arches and close-loop shapes as the main driven force using in this method is gravitational force. In addition, the GDS method take advantage of the thermal-induced creep effect for shaping, so the final geometry of the GDS-fabricated piezoceramic is limited by the surface tension of the liquid-like semifinal specimen for example square and cone shapes.

Table 1 | Comparison of processing methods for fabricating compact piezoceramics with complex geometries by the GDS process, 3D printing and post-processing methods.

	Scalability for scale-up production	Limitations	Compactness [piezoceramic ratio in green bodies]*	Capacity for forming complex geometry	Piezoelectricity [d_{33} pC/N]
3D Printing:					
Slurry-based ^{20,21,23}	Low*	High viscosity requirement on the feedstock, Weak mechanical quality	Low [up to 35 vol%]	Yes	Medium [39-345]
Bulk solid-based ^{28,29,30}	Low*	High viscosity requirement on the molten feedstock, Weak mechanical quality	Low [up to 29 vol%]	Yes	Low [8.72]
Powder-based ^{24,25, 26}	Low*	Not suitable for fabricating low-porosity components, Weak mechanical quality	Low [-]	Yes	Low [2.1]
Post processing:					
Thermal-based ^{31,32,33,34}	Medium*	Not suitable for non-laminate structures, limited applicability on various materials	High [-]	No	Medium [~390]
Pre-stressed ^{35,36}	Medium*		High [-]	No	Medium [~390]
GDS process:	High*	Geometrical limits*, Not suitable for buried sintering	High [99 vol%]	Yes	High [~595]

REVIEWERS' COMMENTS

Reviewer #1 (Remarks to the Author):

This paper can be accepted in the current form.

Reviewer #2 (Remarks to the Author):

I am satisfied with the authors' responses to my comments. I would recommend acceptance of the current version of manuscript for publication.

Reviewer #3 (Remarks to the Author):

Good clear comments and reply to referees. I am happy with the response with the addition of the processing constraints rather than 'None'.

Chris Bowen